# Quantitative Optimization of Handheld Probe External Pressure on Dermatological Microvasculature Using Optical Coherence Tomography-Based Angiography

**DOI:** 10.3390/mi15091128

**Published:** 2024-09-04

**Authors:** Jiacheng Gu, Jinpeng Liao, Tianyu Zhang, Yilong Zhang, Zhihong Huang, Chunhui Li

**Affiliations:** 1School of Physics, Engineering and Technology, University of York, York YO10 5DD, UK; xgq516@york.ac.uk (J.G.); tqq517@york.ac.uk (J.L.); t.x.zhang@dundee.ac.uk (T.Z.); zhihong.huang@york.ac.uk (Z.H.); 2Centre for Medical Engineering and Technology (CMET), School of Science and Engineering, University of Dundee, Dundee DD1 4HN, UK; y.z.z.z.h.zhang@dundee.ac.uk

**Keywords:** OCTA, vessel density, external pressure, dermatology

## Abstract

Optical Coherence Tomography (OCT)-based angiography (OCTA) is a high-resolution, high-speed, and non-invasive imaging method that can provide vascular mapping of subcutaneous tissue up to approximately 2 mm. In dermatology applications of OCTA, handheld probes are always designed with a piece of transparent but solid contact window placed at the end of the probe to directly contact the skin for achieving better focusing between the light source and the tissue, reducing noise caused by minor movements. The pressure between the contact window and the skin is usually uncontrollable, and high external pressure affects the quality of microvascular imaging by compressing the vessels and obstructing the underlying blood flow. Therefore, it is necessary to determine a pressure range to ensure that the vessels can be fully imaged in high-quality images. In this paper, two pressure sensors were added to the existing handheld OCT probe, and the imaging probe was fixed to a metal stand and adjusted vertically to change the pressure between the probe and the tested skin site, a gradient of roughly 4 kPa (with 1–2 kPa error) increase was applied in each experiment, and the impact of pressure to the vessel was calculated. The experiment involved a total of five subjects, three areas of which were scanned (palm, back of the hand, and forearm). The vessel density was calculated to evaluate the impact of external pressure on angiography. In addition, PSNR was calculated to ensure that the quality of different tests was at a similar level. The angiography showed the highest density (about 10%) when the pressure between the contact window on the probe and the test area was between 3 and 5 kPa. As the pressure increased, the vascular density decreased, and the rate of decrease varied in different test areas. After fitting all the data points according to the different sites, the slope of the fitted line, i.e., the rate of decrease in density per unit value of pressure, was found to be 4.05% at the palm site, 6.93% at the back of the hand, and 4.55% at the forearm site. This experiment demonstrates that the pressure between the skin and contact window is a significant parameter that cannot be ignored. It is recommended that in future OCTA data collection processes and probe designs, the impact of pressure on the experiment be considered.

## 1. Introduction

The structure and function of blood vessels are key factors in maintaining skin homeostasis [1]. Changes in the structure and function of blood vessels are significant indicators of diseases. Therefore, effectively imaging blood vessels and analyzing microvascular parameters can identify important biomarkers for skin disease diagnoses, e.g., diabetes mellitus (DM) [2,3,4], arteriosclerosis [5,6] and widespread peripheral vascular diseases [7]. 

Optical Coherence Tomography (OCT) is an advanced non-invasive real-time imaging technology that can provide cross-sectional images with a resolution of up to 5 μm and that has a detection depth of up to 2–3 mm below biological tissues [8]. OCT-based angiography (OCTA) detects changes in tissue reflectance signals caused by the movement of red blood cells [9]. Since the components within blood vessels, such as red blood cells, are constantly moving while other tissues remain relatively stationary, comparing the flow signals through the motion of red blood cells with the light signals from static tissues makes it is possible to differentiate the moving red blood cells within the relatively static tissues of the body, reflecting the structural characteristics of skin blood vessels [2]. Assessing the vascular system provides important support for the diagnosis and monitoring of various skin diseases [10]. In dermatology, it allows for detailed observation of the vascular characteristics of healthy skin, benign skin lesions, psoriasis, scleroderma, and other pathological conditions [11]. 

In the process of using OCTA for skin vascular imaging, the typical probe design usually includes a contact window at the bottom of the probe [12,13,14] to reduce motion artifacts caused by slight movements of the test area and to ensure proper focus between the lens and the test area in tight contact. The pressure applied between the contact window and the test area is a significant operational variable, and the external pressure can indeed affect the vessel mapping. High pressure can directly compress the skin and subcutaneous vessels, leading to reduced or interrupted blood flow [14], thereby reducing the vascular filling and signal intensity in the imaging area and causing inaccurate OCTA quantitative assessment. Therefore, it is essential to determine an optimal pressure range between the contact window and the test area during testing to ensure better imaging for quality and accurate assessments.

This study aimed to demonstrate the effects of pressure applied externally on the skin, vasculature mapping, and OCTA, allowing for the direct imaging of the vasculature at the test site and calculation of the vascular density. This enables the determination of an appropriate range of pressure to be applied during testing to ensure imaging accuracy and maintain real vascular density, and we believe this paper can be used as a reference for future probe design.

## 2. Materials and Methods

In our experiment, there were a total of 5 healthy participants aged between 20 and 30 years (1 female and 4 males). This study was reviewed and approved by the Research Ethics Committee of the University of Dundee (UOD-SSREC-RPG-BioEng-2022-003). The test areas were the thenar eminence of the palm, back of the hand, and outer forearm. Three successive acquisitions were taken from each site under each pressure and the best one was chosen. All participants provided informed consent prior to entering the laboratory for data collection, which also conformed to the tenets of the Declaration of Helsinki.

### 2.1. Experimental Setup

The imaging system used in this study was a lab-built, portable Swept-Source OCT (SSOCT) system equipped with a handheld probe, as shown in Figure 1. This system is a non-invasive, high-resolution, high-speed interferometric imaging device. The system has been well described in our previous papers [12,13,15,16].The system uses a 400 kHz line scanning rate, 1300 nm wavelength, and 100 nm bandwidth swept laser, providing 7.4 μm axial resolution in tissue. The laser beam was split by a beam splitter into reference and sample arms. In the reference arm, an electro-optical delay line and a polarization controller were incorporated for optimal performance, and the sample arm was part of a handheld probe. The scanning ranges in this study were set at 5.2 × 5.2 mm^2^ in both width and length. Each data acquisition involved 400 B-scans per acquisition, and each B-scan consists of 400 A-lines, covering a 2.0 mm depth. The whole probe had a length of 160 mm, height of 158 mm, and width of 148.4 mm, and the contact window under the probe had a diameter of 15 mm.

The probe was mounted on a metal stand that could be moved vertically. The amount of pressure applied was determined by the height at which the stand held the probe, and the pressure could be adjusted by moving the stand up and down. This setup minimized motion artifacts at the operator’s level. 

The pressure sensors (PPS UK Limited, Glasgow, UK, Model PN4290) contained 12 pressure-sensing units, with each unit measuring 2.5 mm × 2.5 mm, and each unit could show the pressure values. However, due to the larger size of the sensors compared to the OCTA probe, only a part of the sensor could fully come into contact the area between the contact window and the test site. Therefore, only a few selected units came completely into contact with the test site. Additionally, a 3D-printed external shield designed using SolidWorks 2022 was added to the existing OCTA contact probe around the contact window. The purpose of this shield was to ensure that the sensor and the contact window were on the same level, as shown in Figure 2. Two pressure sensors were attached to both sides of the shield and connected to a feedback detection system. The pressure value was taken as the average of the feedback from the two symmetrical sensors.

Starting from the point where the contact window just makes contact with the test area, taking individual differences into account, the initial pressure range was controlled to be small (<5 kPa). Subsequently, the pressure intensity between the contact window and the test area was gradually increased from the initial value in step increments of approximately 4 kPa until the participants felt a noticeable pressure at the test site, at a value of about 25 kPa.

### 2.2. Image Processing and Metrics Evaluation

Based on the knowledge of skin layer separation [11,17,18,19], the skin is divided into the epidermis, papillary dermis, and reticular dermis, as shown in Figure 3.

#### 2.2.1. Calculation of Vessel Density

To quantitatively analyze vascular density, the acquired images underwent the following preprocessing steps. First, the input images were resized to 400 × 400 pixels, and the grayscale values were normalized to a range between 0 and 255. Next, a Hessian filter was applied to extract the vascular structures [20]. The filtered results were then subjected to median filtering and threshold processing to remove noise and small areas, resulting in a binarized vascular image.

After preprocessing, the binary images were further analyzed. First, the actual physical size (μm/pixel) corresponding to each pixel in the image was calculated. Then, morphological operations were used to remove small holes and isolated pixels, resulting in a cleaner binary image. The vascular density was calculated, which is the ratio of the vascular pixel area to the total pixel area. This process is illustrated in Figure 4.

#### 2.2.2. Calculation of PSNR

Increasing the number of repeated scans can improve the quality of OCTA images [21]; however, a long scanning time inevitably increases the likelihood of unpredictable movements by the subjects, and a longer collection time (12 repetitions, ~9 s) could cause an unstable pressure between the skin and the contact window in this study. In contrast, a shorter collection time for four repetitions (~3 s) allows the subject to maintain a more stable pressure between the skin and the contact window. Thus, to ensure image quality, four repetitions of scanning were performed. The Peak Signal-to-Noise Ratio (PSNR) was additionally calculated to ensure that the image quality from the low-repetition scanning was comparable with high-repetition scanning images and maintained an acceptable image quality [15].

Peak Signal-to-Noise Ratio (PSNR) is a widely used tool for assessing image quality [22,23]. It is a measure of the ratio of the maximum power of the signal to the power of the noise. PSNR is measured by calculating the Mean Square Error (MSE) between the original and the distorted image and then converting it to a logarithmic scale or the Signal-to-Noise-Ratio (SNR), and this measure is necessary for comparing different reconstructed images. The higher the PSNR value, the more similar the images are, and this experiment considers a PSNR value > 10 to be a usable result. By comparing images with PSNR > 10 and <10, as shown in Figure 5 below, there are significant differences between images with a PSNR < 10. 

The equation for PSNR is shown in Equation (1):(1)$$PSNR=10{log}_{10}\left(\frac{I_{max}^{2}}{MSE\left(I,\hat{I}\right)}\right)$$
where I is the reference image (12-repeat), Î is the image from collection (4-repeat), and I_max_ is the maximum possible pixel value of the image. MSE is the Mean Square Error.

## 3. Results

In this experiment, vascular imaging was performed on different areas under various pressures, as shown in Figure 6 and Figure 7. Each area was divided into a superficial layer of the papillary dermis and a deep layer of the reticular dermis for enface imaging, and all the PSNR calculation results from this experiment were >10.

The data obtained were then summarized and the initial average pressure values were recorded, and decreases in vascular density and their standard deviations for the different test sites were calculated. The average initial pressure for the palm area was 4.62 kPa, for the back of the hand it was 3.96 kPa, and for the forearm it was 4.92 kPa. Subsequently, the vascular density of each image was calculated using the previously described method. Based on the trends in density decrease, decline rate curves were plotted according to the different test sites, and a linear regression was also fitted to the data using the least squares method for all data. The results show that after several increases in pressure at different test sites, the average vascular density in the superficial layer, deep layer, and their combined layer showed a decreasing trend, as illustrated in Figure 8.

After fitting, the slopes of the fitted lines and the R^2^ were calculated at the different sites. The slope represents the rate of decrease in density per unit of pressure, and the R^2^ ensures that the fitted data have a strong explanatory power. The results show that for each unit increase in pressure, the overall vessel density decreased by 4.05% at the palm site, 6.93% at the back of hand site, and 4.55% at the forearm site. In addition, the above operations were also carried out in all the superficial and deep layers (papillary dermis and reticular dermis) of the different test sites, and the data are shown in Table 1.

Here, a negative value of slope represents the tendency of the vessel density to decrease when increasing the pressure, and the value of slope represents the value of the vessel density decreasing as a percentage when the pressure is increased by every 1 kPa; all the R2 values tend to 1, which also proves that the fit is scientifically valid.

## 4. Discussion

The experiment demonstrates that as the external pressure between the contact window beneath the probe and the test area increases, it can cause interruptions in blood flow, leading to reduced vascular density in the imaging and affecting the imaging quality. In this experiment, by adding a pressure sensor under the contact window, a suitable pressure range was identified, within the range of 3–5 kPa, and real imaging results were achieved. It was found that for each unit increase in the pressure value, the overall vessel density decreased by 4.05% at the palm site, 6.93% at the back of hand site, and 4.55% at the forearm site. Furthermore, when dividing each area into superficial and deep vessel layers according to the papillary dermis and reticular dermis, it was observed that increased pressure had a more significant impact on the superficial vessel layer.

It is worth noting that at the back of hand site, vascular density decreases somewhat more rapidly with increasing external pressure, probably because of the thinner skin thickness at the back of hand site. However, the results of the experiment could not prove that thinner areas of the skin (back of the hands, forearm) had a more significant effect than thicker areas (palm). This may be related to the distribution and morphology of the blood vessels at the collection sites [24], and it is also related to the gender of the participant, their skin condition, and the differences in skin thickness and vascular distribution at the same location between each individual. Moreover, when the contact window first contacts the skin, it is difficult to control it to the same initial pressure due to the varying skin and physiological conditions of each participant [25,26]. When analyzing vascular density, it was found that the decrease rate across the two layers varies across different sites. This may be due to the thickness of the epidermis or the presence of elastic substances in the subcutaneous tissue, such as elastic fibers, collagen, and adipose tissue [27], which may also potentially affect the accuracy of the pressure sensor’s readings.

This paper will be useful to guide others in OCT/A probe design. Based on the findings, it is recommended to add pressure sensors on both sides of the contact window or other nearby areas. This will allow for control of the pressure within the precise range suggested in this paper, in order to obtain an accurate depiction of the vascular distribution.

## 5. Conclusions

This experiment demonstrates the effects of externally applied pressure on skin vasculature, and OCTA allows for direct imaging of the vasculature at the test site and calculation of the vascular density. This enables the determination of an appropriate range of pressure (3–5 kPa) to be applied during testing to ensure that imaging is real and maintains good vascular density. Further, in future OCT/A data collection processes and probe design, the impact of pressure on the experiment should be considered as throughout this paper.

## Figures and Tables

**Figure 1 micromachines-15-01128-f001:**
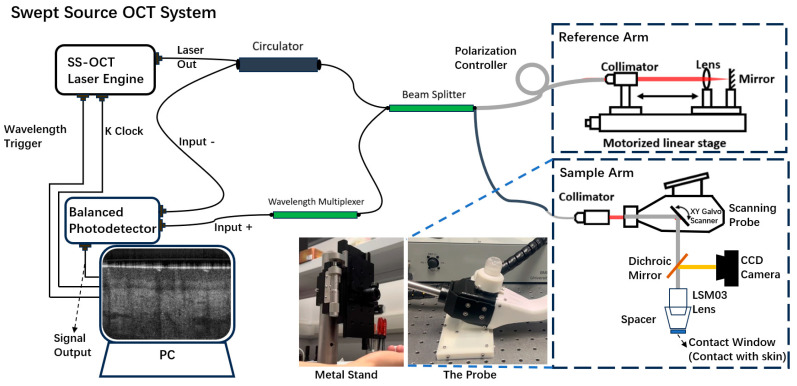
Schematic diagram of the Swept-Source Optical Coherence Tomography (SSOCT) system.

**Figure 2 micromachines-15-01128-f002:**
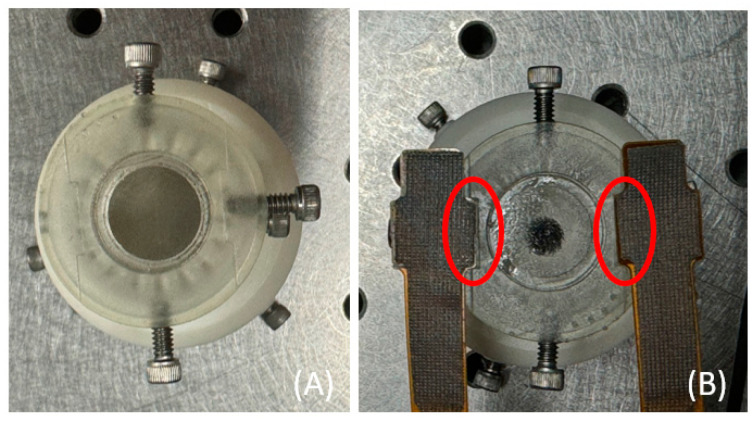
An additional external shield (**A**) was added to the probe, which could accommodate two sensors on the opposite side, as shown in (**B**). Due to their large size, only a part of each sensor (within the red circles) could fully come into contact between the contact window and the test area.

**Figure 3 micromachines-15-01128-f003:**
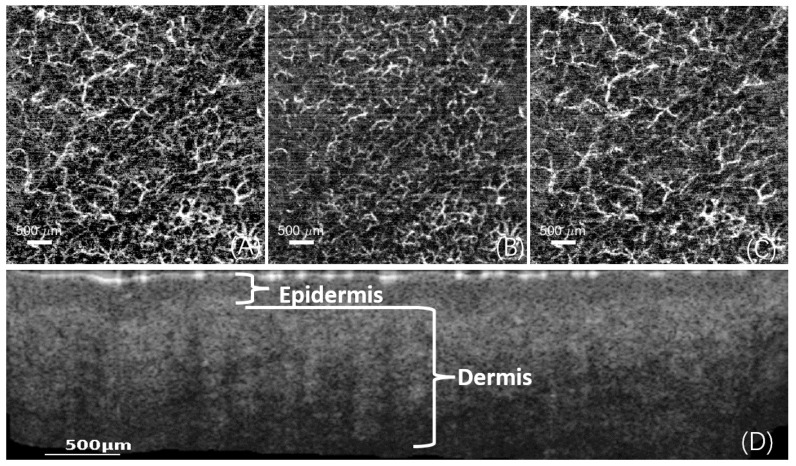
Layer separation, including an enface image containing all depth information (**A**). After dividing the subcutaneous layers into three parts, it shows the enface image within the papillary dermis (**B**), the enface image within the reticular dermis (**C**), and the layered structure of the skin (**D**).

**Figure 4 micromachines-15-01128-f004:**
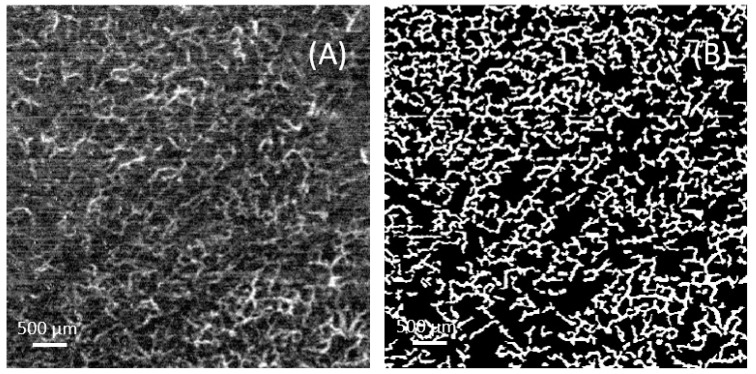
An example for calculation of the vessel density: (**A**) the grayscale image; (**B**) the binarized image where the white areas represent the blood vessels. The final vascular density is the ratio of the white area to the total area.

**Figure 5 micromachines-15-01128-f005:**
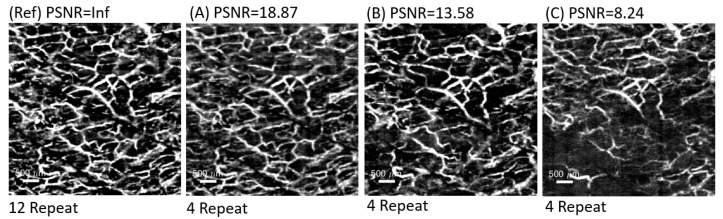
PSNR comparison images; (Ref) represents the reference image used for calculating the PSNR, while (**A**–**C**) represent cases where the PSNR value is >15, >10, and <10.

**Figure 6 micromachines-15-01128-f006:**
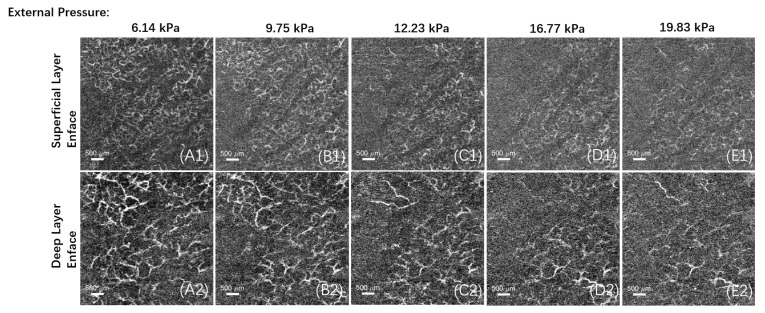
Enface images of different layers, from (**A**–**E**), the pressure intensity between the contact window and the test area increased in steps of approximately 4 kPa, the subfigure (**1**) and (**2**) represent the superficial vessels in the papillary dermis and the deep vessels in the reticular dermis, respectively. In this example, the initial pressure value is 6.14 kPa, with an initial vascular density of 6.4%. After four additional pressure step increments, the pressure value between the contact window and the test area is 19.83 kPa, and the vascular density decreases to 2.9%.

**Figure 7 micromachines-15-01128-f007:**
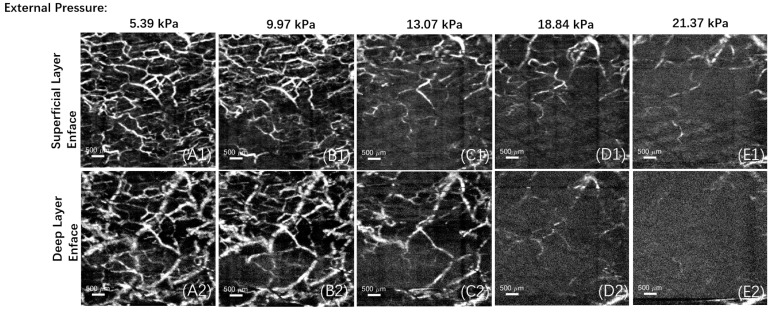
Enface images of different layers, from (**A**–**E**), the pressure intensity between the contact window and the test area increased in steps of approximately 4 kPa, the subfigure (**1**) and (**2**) represent the superficial vessels in the papillary dermis and the deep vessels in the reticular dermis, respectively. In this example, the initial pressure value is 5.39 kPa, with an initial vascular density of 24.1%. After four additional pressure step increments, the pressure value between the contact window and the test area is 21.37 kPa, and the vascular density decreases to 3.5%.

**Figure 8 micromachines-15-01128-f008:**
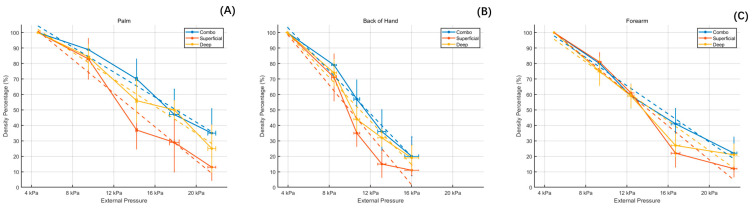
The trend in vascular density reduction and the rate of decrease curve under different pressures for different test sites, the dotted lines are the fitted lines; (**A**) represents the palm, (**B**) represents the back of the hand, and (**C**) represents the forearm.

**Table 1 micromachines-15-01128-t001:** The slopes of the fitted data for the different sites and the R^2^.

		Slope (Percentage per kPa)	R^2^
Palm	Combo	−4.05	0.975
	Superficial	−5.45	0.958
	Deep	−4.37	0.977
Back of the hand	Combo	−6.93	0.982
	Superficial	−8.06	0.939
	Deep	−7.04	0.969
Forearm	Combo	−4.55	0.985
	Superficial	−5.46	0.953
	Deep	−4.77	0.943

## Data Availability

The data presented in this study are available on request from the corresponding author. The data are not publicly available due to ethical restrictions.

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
