# Peer review of "Quantitative Optimization of Handheld Probe External Pressure on Dermatological Microvasculature Using Optical Coherence Tomography-Based Angiography"

_micromachines, 2024, doi:10.3390/mi15091128_

Round 1

Reviewer 1 Report

Comments and Suggestions for Authors

General Comments:

The manuscript presents a study that addresses an important technical challenge in the application of OCT-based angiography in dermatology. The authors have focused on optimizing the external pressure applied by a handheld probe to improve the quality of microvascular imaging, which is a critical factor in obtaining reliable and reproducible results. The study tackles a relevant issue in the field of dermatological imaging. The optimization of probe pressure to enhance image quality and prevent vascular compression is a novel approach that could have practical implications for future OCTA device design and usage. The manuscript effectively presents quantitative data regarding the impact of pressure on vascular density, which is critical for validating the findings. The differentiation of results based on different test areas adds depth to the analysis.

Overall, the manuscript presents significant findings with practical relevance in dermatological imaging using OCTA. However, there are a few points that require further clarification and improvement. I recommend the acceptance of this work with major revisions, provided that the following comments and questions are fully addressed.

Specific Comments:

1.       The role of PSNR in the research should be clarified, especially how it can be used to assess and ensure consistency across test qualities.

2.       The methods used for pressure control and the criteria for selecting the pressure range should be mentioned in the abstract. This will provide readers with a clearer understanding of the experimental procedure and the basis for the conclusions drawn.

3.       It is noted that the manuscript reports the highest vascular density (about 10%) when the pressure between the contact window on the probe and the test area is between 3-5 kPa. However, it is unclear why the highest density is observed at this pressure range, as one might expect the highest density to be achieved at zero pressure or when there is no contact. Clarification is needed on why vascular density is highest at this specific pressure range rather than at zero pressure.

4.       It is known that OCTA is sensitive to motion artifacts. It should be addressed whether different pressures applied by the hand-held probe affect motion artifacts, and whether higher pressure results in increased probe stability.

5.       The manuscript mentions determining an appropriate range of pressure. It is important to clarify how the pressure range is precisely controlled during hand-held imaging. Is the pressure control achieved solely through manual application of force, or are there additional mechanisms or techniques used to ensure accuracy?

6.       The scan range is 5.2 × 5.2 mm, with 400 × 400 A-lines collected. Therefore, the imaging results should be in undersampling. Would more spatial sampling points give a better result?.

7.       Some necessary detailed information of the probe is missing, including the probe size and imaging speed.

8.       The manuscript notes that due to the larger size of the sensors compared to the OCTA probe, only a portion of the sensor can fully contact the area between the contact window and the test site. It is important to address whether this partial contact affects the accuracy of the pressure measurements.

9.       The scale bar in Figure 3 is unclear. It is recommended to redraw and clarify the scale bar to ensure that it is easily readable and accurately represents the scale of the image.

10.    The layout and format of figures 5 and 8 should be improved. It is recommended that these figures be repositioned and designed to be centered.

11.    It is important to address whether there are mechanisms in place to handle the issue of pressure instability caused by hand tremor during imaging. Clarification on how this potential source of variability is managed would be beneficial.

12.    Attention should be given to formatting errors throughout the manuscript. For example, 'The scanning ranges in this study were set as 5.2 × 5.2 mm2 in both width and length' should be correctly formatted as '5.2 × 5.2 mm²'. A thorough review of the manuscript for similar issues is recommended.

Comments on the Quality of English Language

Minor editing of English language required

Reviewer 2 Report

Comments and Suggestions for Authors

This manuscript reports about changes in microvasculature in OCTA image according to applied pressure. Although this is an important issue in clinical research, the study design is not rigorous enough and there are not much insight to gain from this work.

Figure 8 and Table 1 seem to be the most important result, but they are far from being useful to scientific readers. Pressure, which is the controlled variable in this study, is very poorly controled (in 3-5kPa step), and the decreasing ratio of vascular density is very roughly determined. If it turns out to be difficult to control external pressure, the x-coordinates of datapoints in Fig 8 should have been positioned at whatever pressure value the pressure sensor reads, avoiding the use of categorical x-axis. Otherwise, the slope calculated from this graph is meaningless. The numbers in Table 1 also has to have units in [% per kPa], not just [%].

Also be specific about how authors calculated the tolerance value that appears in Table 1. If linear regression has been used, there has to be reasons as to why the density percentage is believed to behave linearly in terms of external pressure. Also, it's not clear whether authors have used just least square method or weighted least square method.

Additionally, authors doesn't seem to explain the meaning of those slope values in Table 1 very well. What kind of hypothesis do authors have about the pressure-dependence of vasculature in different type of tissues, and how does this experimental result prove/disprove it?

Comments on the Quality of English Language

Some minor flaws detected in English usage, such as in line 99.

Round 2

Reviewer 1 Report

Comments and Suggestions for Authors

The author responded to all reviewers' concerns and made appropriate corrections to the article. No further comments or suggestions.

Comments on the Quality of English Language

 Moderate editing of English language required.

Author Response

Thank you for your time and all your constructive suggestions!

Reviewer 2 Report

Comments and Suggestions for Authors

There has been some revisions made in the right direction, but the manuscript still lacks scientific soundness. Let's take examples from Fig 8 and Table 1.

It is understood that pressure sensor reading fluctuates a lot, which is common in real world setting, but the way authors treated these pressure data doesn't seem to be scientifically rigorous. In Fig 8, each datapoint has to have errorbars not just in y-direction but also in x-direction, as authors have admitted that pressure readings fluctuated significantly. And the x-direction errorbars should be different for all datapoints if authors have actually measured it. 

Also, in Table 1, authors said they calculated 'average decrease' in different layers without making any linear regression, but what does that exactly mean? If it means that authors have calculated slope between each consecutive datapoints, which results in 4 different slopes, and just took average of them, how can it represent the total behavior of each different layers? If we look at, let's say, Palm for Superficial Layer, the average decrease is 29.2% per 4 kPa. Then the density percentage at 20 kPa should be lower than 0 %, which doesn't make sense. Shouldn't it be more appropriate to use linear regression to get the slope and R^2 value for each layer?

Also, the unit of slope in table 1 has to be [% per kPa], not [% per 3-5kPa], meaning the numbers has to be recalculated.

Overall, it is advised that authors perform a more appropriate analysis on the data shown in Fig 8 and draw more scientific conclusions out of it. The current state of the manuscript doesn't seem to be acceptable for publishing.

Round 3

Reviewer 2 Report

Comments and Suggestions for Authors

Authors' revision looks alright, except that the slope values in Table 1 has to have a specific unit. No further comments.